# Impairment of visually guided associative learning in children with Tourette syndrome

**Gabriella Eördegh**[1]☯, **Ákos Pertich**[2]☯, **Zsanett Tárnok**[3], **Péter Nagy**[3], **Balázs Bodosi**[2], **Zsófia Giricz**[2], **Orsolya Hegedűs**[3], **Dóra Merkl**[3], **Diána Nyujtó**[2], **Szabina Oláh**[3], **Attila Őze**[2], **Réka Vidomusz**[3], **Attila Nagy**[2]*

**1** Faculty of Health Sciences and Social Studies, University of Szeged, Szeged, Hungary, **2** Department of Physiology, Faculty of Medicine, University of Szeged, Szeged, Hungary, **3** Vadaskert Child and Adolescent Psychiatry, Budapest, Hungary

☯ These authors contributed equally to this work.
* nagy.attila.1@med.u-szeged.hu

## Abstract

The major symptoms of Tourette syndrome are motor and vocal tics, but Tourette syndrome is occasionally associated with cognitive alterations as well. Although Tourette syndrome does not affect the majority of cognitive functions, some of them improve. There is scarce evidence on the impairment of learning functions in patients with Tourette syndrome. The core symptoms of Tourette syndrome are related to dysfunction of the basal ganglia and the frontostriatal loops. Acquired equivalence learning is a kind of associative learning that is related to the basal ganglia and the hippocampi. The modified Rutgers Acquired Equivalence Test was used in the present study to observe the associative learning function of patients with Tourette syndrome. The cognitive learning task can be divided into two main phases: the acquisition and test phases. The latter is further divided into two parts: retrieval and generalization. The acquisition phase of the associative learning test, which mainly depends on the function of the basal ganglia, was affected in the entire patient group, which included patients with Tourette syndrome with attention deficit hyperactivity disorder, obsessive compulsive disorder, autism spectrum disorder, or no comorbidities. Patients with Tourette syndrome performed worse in building associations. However, the retrieval and generalization parts of the test phase, which primarily depend on the function of the hippocampus, were not worsened by Tourette syndrome.

## Introduction

Tourette syndrome (TS) is a disorder that presents before the age of 18 years, affecting 1% of school-aged children [1–4]. The most frequent symptoms are motor and vocal tics [1, 5], which significantly improve in many patients by young adulthood [6]. In addition to these primary symptoms, pure Tourette syndrome is associated with some mild alterations in cognitive functions, mainly in a few executive functions (i.e., verbal fluency, working memory, and Stroop effect), which extend into adulthood [7–9], and others, which disappear with age (i.e., deficits demonstrated by the Wisconsin Card Sorting Test [10]).

**Data Availability Statement:** All relevant data are within the manuscript and its Supporting Information files.

**Funding:** (AN) Faculty Research Fund, Albert Szent-Györgyi Grant, Faculty of Medicine,

University of Szeged. SZTE ÁOK-KKA grant No:
2019/270-62-2. http://www.med.u-szeged.hu/
karunkrol/kari-palyazatok/aok-kari-kutatasi-alap-
181005 The funders did not play any role in the
study design, data collection and analysis, decision
to publish, or preparation of the manuscript.

**Competing interests:** The authors have declared
that no competing interests exist.

The symptoms of Tourette syndrome are mainly related to dysfunction of the basal ganglia and the connected frontal lobe [11, 12]. Reduced left caudate nucleus volume [13], prefrontal hypertrophy, and structural changes have been described in Tourette syndrome [14–16]. The connection between the frontal lobe and the basal ganglia via parallel and overlapping frontostriatal circuits [5, 12, 17–22] is significantly weaker in Tourette syndrome [23].

This frontostriatal system is responsible for motor functions and several cognitive functions [24]. However, significant impairment of cognitive functions has only been rarely described in patients with Tourette syndrome without any comorbidities [25], and the impairment often depends on the level of tic severity [26–28]. Most impairment has been reported in Tourette syndrome with its most frequent comorbidity, attention deficit hyperactivity disorder (ADHD) [29–31]. Previous studies have emphasized that most alterations of cognitive functions are primarily associated with concomitant ADHD and TS [32–36]. These results suggest that the cognitive performance of patients with TS + ADHD is more similar to that of patients with ADHD than that of patients with Tourette syndrome [33, 35, 37, 38]. Accordingly, Channon et al. did not find any impairment in explicit or implicit memory or learning processes in Tourette syndrome alone but did find these impairments in TS + ADHD [39]. In reinforcement learning, the results are conflicting, but most results show no difference between patients with Tourette syndrome and healthy controls [26, 40–43]. Patients with Tourette syndrome have intact motor sequence learning [44], but the procedural (habit) learning in a probabilistic classification learning, which is connected to the dorsal striatum [45] was significantly altered [46, 47]. However, hippocampus-related learning was not affected in patients with Tourette syndrome alone [47]. Another procedural learning type, implicit probabilistic sequence learning, was not affected or was even better in patients with Tourette syndrome [48, 49]. These learning functions function via frontostriatal loops as well as associative learning, which has not yet been investigated for Tourette syndrome.

Associative learning, in which discrete and often different signals are linked together, is a type of conditioning. For example, when we remember a face, we record all the facial features, and the parts reinforce each other. This basic cognitive function is related to basal ganglia and hippocampus functions. The Rutgers Acquired Equivalence Test [35] investigates this specific learning ability. The primary advantage of this test is that each phase of the paradigm has well-described neural substrates. The acquisition phase, which primarily depends on the function of the basal ganglia [35, 50], tests the association of two different visual stimuli with the help of feedback about the correctness of the responses. In the test phase, which primarily depends on the function of the hippocampus and the mediotemporal lobe [35, 50], the previously learned associations are presented without any feedback (retrieval part), and previously not presented but predictable associations (generalization part) are shown. This learning function was previously investigated in adult patients with Parkinson's disease, Alzheimer's disease, schizophrenia, and migraine without aura [35, 51–53] but never in children with neurological or psychiatric disorders compared with healthy controls. Thus, the description of this learning ability in Tourette syndrome remains missing. Since Tourette syndrome is related to dysfunction of the basal ganglia and the frontostriatal loops, we hypothesized that the acquisition phase could be primarily affected in the Acquired Equivalence Test. Thus, the primary aim of the present study was to determine whether visually guided associative acquired equivalence learning is affected in children with Tourette syndrome. We also investigated whether similar to other cognitive deficits of patients with Tourette syndrome the ADHD is the primary reason for reduced associative learning ability in patients with Tourette syndrome.

## Methods

### Participants

Altogether, 46 children with Tourette syndrome participated in the present research. The children were recruited from Vadaskert Child Psychiatry Hospital in Budapest, Hungary. The children were diagnosed by both a licensed clinical psychologist and a board-certified child psychiatrist at the hospital according to the Diagnostic and Statistical Manual of Mental Disorders, Fifth Edition (DSM-V) criteria [1]. A total of 21 patients were diagnosed with Tourette syndrome without any other neurological or psychiatric comorbidities (TS group); 15 were diagnosed with Tourette syndrome and comorbid ADHD (TS + ADHD group); and 10 were diagnosed with Tourette syndrome and some other comorbidity (obsessive compulsive disorder [OCD] or autism spectrum disorder [ASD]; TS + OCD/ASD group). In this study, we analyzed associative learning of patients in the TS, TS + ADHD, and TS + OCD/ASD groups in detail (32 boys and 14 girls, mean age: 11.64±2.38 years, age range: 8–17 years). Children with other neurodevelopmental or psychiatric comorbidities or learning disabilities were excluded. The mean Yale Global Tic Severity Scale (YGTSS) total tic score (TTS) was 20.7±6.4 (range: 8–33) [5, 54] in the whole patient group. Two participants showed minimal tic severity (TTS ≤ 10); 17 showed mild tic severity (score 11–20); and 19 showed moderate to severe tic severity (score > 20). There were no significant differences (Kruskal–Wallis ANOVA, p>0.05) among the patient subgroups according to age, IQ level, and tic severity. Twelve of the children involved in this study (3 from the TS group, 6 from the TS + ADHD group, and 3 from the TS + OCD/ASD group) were medicated because of the symptoms of their disorder. The TS group received dopamine 2 receptor antagonists (haloperidol and risperidone). The TS + ADHD patients received a norepinephrine–dopamine reuptake inhibitor (methylphenidate), a dopamine 2 receptor antagonist (haloperidol), or a partial agonist of the dopamine 2 and serotonin 1A receptors (aripiprazole), a norepinephrine transporter and dopamine reuptake inhibitor (atomoxetine), or melatonin. The TS + OCD/ASD group received selective serotonin reuptake inhibitors (fluvoxamine and sertraline), a partial agonist of the dopamine 2 and serotonin 1A receptors (aripiprazole), a serotonin and dopamine antagonist (risperidone), or a norepinephrine-dopamine reuptake inhibitor (methylphenidate).

The parents of all participants signed an informed consent form and did not receive financial compensation for their participation. The protocol of the study conformed to the tenets of the Declaration of Helsinki in all respects, and it was approved by the Ministry of Human Capacities in Budapest, Hungary (11818-6/2017/EÜIG).

From our database of control children recruited from local schools, 46 control children (31 boys and 15 girls, mean age: 11.55±2.38 years, range: 8–17.5 years) were assorted and individually matched based on sex, age (differing in age by no more than six months), and IQ level to the patient groups. There were no significant differences (Kruskal–Wallis ANOVA, p>0.05) among the control subgroups according to age and IQ level. Table 1 shows the demographic data for the patient and control groups.

The control group only included children without any known psychiatric, neurological, or neurodevelopmental disorders. All participants (patients and controls) had normal or corrected-to-normal vision and normal hearing. The intactness of color vision was tested by Ishihara plates prior to testing to exclude color blindness [55] both in the patient and control groups. Only patients and controls with normal color vision were analyzed in the present study. We estimated the IQ level with Raven's Standard [56] and Colored [57] Progressive Matrices [58].

Table 1. Demographic parameters of the investigated groups.

| Group | Number of cases | Male | Age, mean ± SD (years) | Age, range (years) |
|---|---|---|---|---|
| All patients | 46 | 32 | 11.64±2.38 | 8–17 |
| All controls | 46 | 31 | 11.55±2.38 | 8–17.5 |
| TS | 21 | 14 | 11.74±2.26 | 9–17 |
| TS controls | 21 | 13 | 11.50±2.30 | 9–17 |
| TS + ADHD | 15 | 12 | 11.20±2.10 | 9–16.5 |
| TS + ADHD controls | 15 | 12 | 11.27±2.12 | 9–16 |
| TS + OCD/ASD | 10 | 6 | 12.10±2.84 | 8–17 |
| TS + OCD/ASD controls | 10 | 6 | 12.05±2.88 | 8–17.5 |

TS: Tourette syndrome, ADHD: attention deficit hyperactivity disorder, OCD/ASD: obsessive compulsive disorder or autism spectrum disorder.

## Visually guided associative learning paradigm

The principle of the visual learning paradigm is based on the Rutgers Acquired Equivalence Test [35]. The original visual associative learning test [35] written for iOS (Apple Inc.'s operating system) was slightly modified, translated to Hungarian, and rewritten in Assembly (for Windows) with the written permission of Prof. Catherine E. Myers (Rutgers University, NJ, USA). The test was run on a PC. The testing sessions occurred in a dark and quiet room with the participants sitting at a standard distance (114 cm) from the computer screen with comfortable visibility and legible brightness. The participants were asked to learn associations between antecedent stimuli (four faces: A1, A2, B1, and B2) and consequent stimuli (four fish with different colors: X1, X2, Y1, and Y2). The four possible faces were a male adult, a male child, a female adult, and a female child. The four colors were red, green, blue, and yellow. The antecedent-consequent pairings were randomly generated by the computer from these stimuli for each participant. The acquired equivalence paradigm was structured as follows (Fig 1).

**Acquisition phase.** During each trial of the task, participants saw a face and a pair of fish and had to learn through trial and error which of the fish matched which face. In the initial training stages, participants were expected to learn that when face A1 or A2 appears, the correct choice is fish X1 over fish Y1; when face B1 or B2 appears, the correct choice is fish Y1 over fish X1. If the associations are successfully learned, participants also learn that faces A1 and A2 are equivalent with respect to the associated fish (faces B1 and B2 are likewise equivalent with respect to the associated fish). Next, participants learned a new set of pairs: if presented with face A1, they had to choose fish X2 over Y2, and in the case of face B1, fish Y2 over X2. Altogether, six stimulus combinations were shown in the acquisition phase of the paradigm in which the computer provided feedback about the success of the acquisition after each trial. New associations were individually introduced during the acquisition stages. New associations were mixed with trials of previously learned associations. The participants had to achieve a certain number of consecutive correct responses after the presentation of each new association (4 after the presentation of the first association, and 4, 6, 8, 10, and 12 with the introduction of each new association, respectively) to be allowed to proceed. The number of trials in the acquisition phase was not constant. It depended on the performance of the participant in learning the associations.

**Test phase (retrieval and generalization parts).** After successful acquisition, the participant continued with the test phase of the paradigm, in which no more feedback was provided about the correctness of the choices. The participant had to recall the six previously built associations (retrieval part) and had to make two new but predictable associations (generalization part). In the generalization part of the test, the participant was asked to choose fish X2 or Y2

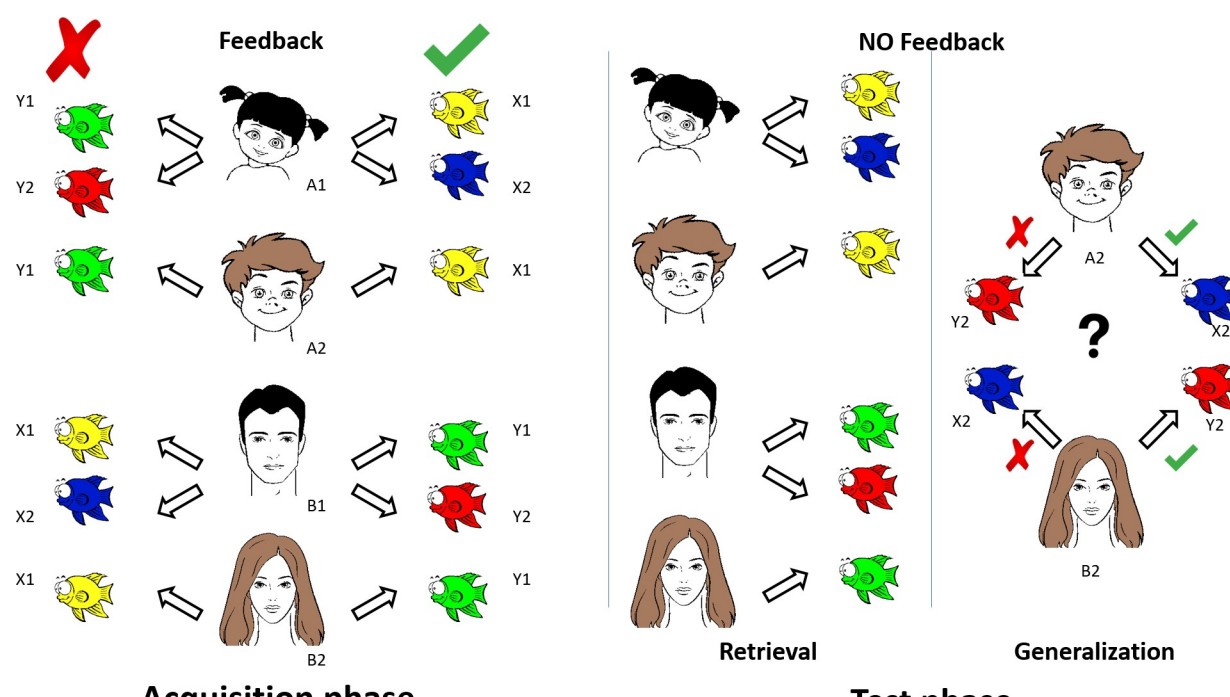

**Fig 1. Graphic overview of the visually guided acquired equivalence learning paradigm.** See details in Methods.

when face A2 or B2 was presented. Having learned that faces A1 and A2 were equivalent in the acquisition phase, participants may generalize from learning that if A1 goes with X2, A2 also goes with X2; the same holds for B2 (equivalent to B1) and Y2 (associated with B1). In the test phase, the new associations were mixed with the previously learned associations. The test phase consistently contained 48 trials, including 36 previously built associations (retrieval part) and 12 new, previously not presented but predictable associations (generalization part). The participants' task throughout the acquisition and testing phases was to indicate their choice in each trial by pressing one of two keyboard buttons labeled LEFT and RIGHT.

Participants were tested individually without a time limit, so they could pay undivided attention to learning. No forced quick responses were expected. While the formal description may imply that the task was difficult, healthy children and intellectually disabled individuals reliably make these kinds of generalizations.

## Data analysis

The number of trials in the acquisition phase and the response accuracy (error ratios) in the acquisition phase, the retrieval part of the test phase, and the generalization part of the test phase were analyzed. We registered the number of trials required to complete the acquisition phase (the number of acquisition trials [NAT]), the number of correct and incorrect choices during the acquisition phase, and the number of correct and incorrect responses for known and unknown associations during the retrieval and generalization parts of the test phase. Using these data, the error ratios were calculated by dividing the number of incorrect responses by the total number of responses provided. The proportion of the number of incorrect responses in the acquisition phase (the acquisition learning error ratio [ALER]), the number of incorrect responses divided by the total number of responses [36] in the retrieval part of the test phase (i.e., the retrieval error ratio [RER]), and the number of incorrect responses

divided by the total number of responses [12] in the generalization part of the test phase (the generalization error ratio [GER]) were measured.

## Statistical analysis

First, we tested the distribution of our data. If the data sets were not normally distributed according to the Shapiro–Wilk normality test, the comparisons between the performance of patients with Tourette syndrome and that of control children were performed with the Mann–Whitney rank test. A Kruskal–Wallis ANOVA was used to compare the performances of the TS, TS + ADHD, and TS + OCD/ASD groups and to compare the performances of the control subgroups, too. The median values and ranges are presented in the results section. If the data were normally distributed according to the Shapiro–Wilk test but the homogeneity of the variance test revealed different variance in the performances of patients with Tourette syndrome and that of healthy control children, Welch's t-test was used to compare the two groups. The mean and SD values are presented in the results section. Statistical analyses were performed in Statistica 13.4.0.14 (1984–2018 TIBCO Software Inc., Palo Alto, CA, USA) and CogStat 1.8.0 and 1.9.0 (2012–2020 Attila Krajcsi).

## Results

In this study, we present the performance of 46 pediatric patients with Tourette syndrome with and without comorbidities and 46 matched healthy control children. All of the participants completed the entire visually guided acquired equivalence learning paradigm.

### The performances of the entire Tourette syndrome group with and without comorbidities versus healthy control children

In order to reduce the effect of multiple (twice in this case) application of the same data, the statistical results were evaluated after Bonferroni correction at a significance level of 0.025. The median NAT was 79.0 (range: 42–202, n = 46) in all patients with Tourette syndrome (with and without medication) and 62.0 (range: 46–124, n = 46) in the control group. The NAT values were significantly higher in patients with Tourette syndrome (Mann–Whitney rank test U = 636, p < 0.001). The median ALER was 0.102 (range: 0–0.325, n = 46) in all patients with Tourette syndrome and 0.085 (range: 0–0.186, n = 46) in the control group. The ALER values, similar to the NAT values, were significantly higher in patients with Tourette syndrome (Mann–Whitney rank test U = 690, p = 0.004). In the retrieval part of the test phase, there was no statistically significant difference (Mann–Whitney rank test U = 1.17e+0.3, p = 0.360) between the patients with Tourette syndrome (median: 0.056, range: 0–0.333, n = 46) and the control group (median: 0.083, range: 0–0.472, n = 46). In the generalization part of the test phase, similar to the retrieval part, there was no statistically significant difference (Mann–Whitney rank test U = 1.26e+0.3, p = 0.103) between the patients with Tourette syndrome (median: 0.125, range: 0–0.667, n = 46) and the control group (median: 0.167, range: 0–0.917, n = 46, Fig 2).

### The effect of medication on the performances of patients with Tourette syndrome with and without comorbidities

To examine the effects of medications on the performances in the applied associative learning test, we compared the performances of the unmedicated patients (TS, TS + ADHD, and TS + OCD/ASD) and their matched healthy controls, the unmedicated patients with Tourette

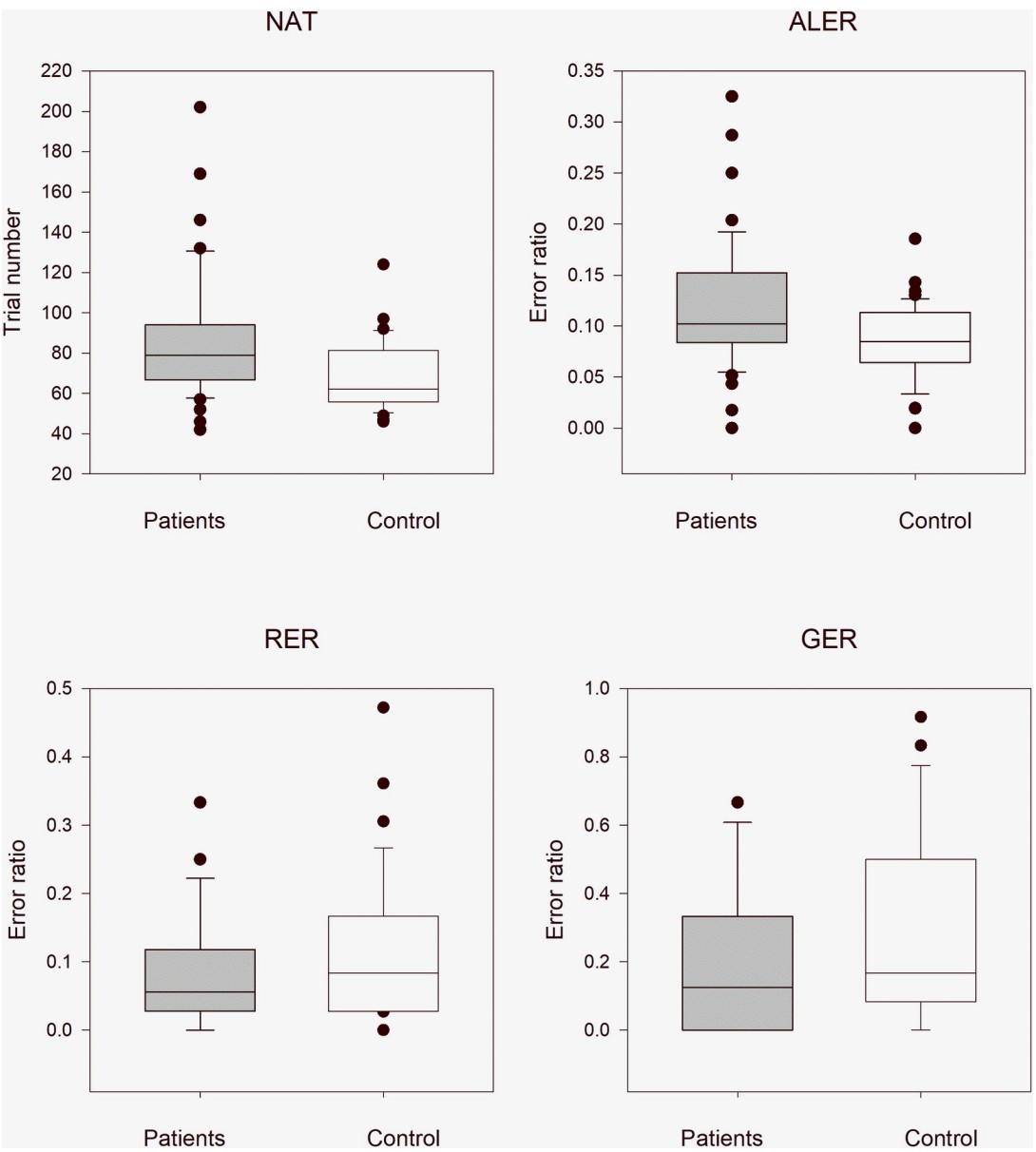

**Fig 2. Performance of all patients with Tourette syndrome and healthy control children in the visually guided equivalence learning paradigm.** NAT denotes the number of the necessary trials in the acquisition phase of the paradigm. ALER shows the error ratios in the acquisition phase of the paradigm. Lower diagrams denote the error ratios in the retrieval (RER) and generalization (GER) parts of the test phase, respectively. In each panel, the first column (gray) shows the performance of all patients with Tourette syndrome, and the second column (white) denotes the performance of the control group. The lower margin of the boxes shows the 25th percentile; the line within the boxes marks the median; and the upper margin of the boxes indicates the 75th percentile. The error bars (whiskers) above and below the boxes indicate the 90th and 10th percentiles, respectively. The dots over and under the whiskers show the extreme outliers. The black stars indicate statistically significant differences (p < 0.05).

syndrome and all patients with TS, and the medicated and unmedicated patients with Tourette syndrome.

**Unmedicated pediatric patients with Tourette syndrome versus healthy control children.** In order to reduce the effect of multiple (twice in this case) application of the same data (first application was above in the comparison with the entire TS group), the statistical results

were evaluated after Bonferroni correction at a significance level of 0.025. The median NAT was 78.5 (range: 42–202, n = 34) in all unmedicated patients and 60.5 (range: 46–124, n = 34) in the matched control group. The NAT values were significantly higher in patients with Tourette syndrome (Mann–Whitney rank test, U = 345, p = 0.004). The median ALER was 0.102 (range: 0–0.325, n = 34) in patients with Tourette syndrome and 0.086 (range: 0–0.186, n = 34) in the control group. The ALER values, similar to NAT values, were significant higher in patients with Tourette syndrome (Mann–Whitney rank test U = 392, p = 0.023). In the retrieval part of the test phase, there was no statistically significant difference (Mann–Whitney rank test U = 657, p = 0.330) between the TS group (median: 0.056, range: 0–0.333) and the control group (median: 0.083, range: 0–0.472). In the generalization part of the test phase, there was no statistically significant difference (Mann–Whitney rank test U = 734, p = 0.053) between the patients with Tourette syndrome (median: 0.083, range: 0–0.667) and the control group (median: 0.208, range: 0–0.917, Fig 3)

**All pediatric patients with Tourette syndrome versus unmedicated patients with Tourette syndrome.**   Comparing the performances of the whole patient group (TS, TS + ADHD, and TS + OCD/ASD) with the unmedicated patient group (TS, TS + ADHD, and TS + OCD/ASD), we did not find any significant differences. The median NAT was 79.0 (range: 42–202, n = 46) in the whole patient group and 78.5 (range: 42–202, n = 34) in the unmedicated patient group. There was no significant difference in the NAT between these groups (Mann–Whitney rank test U = 763, p = 0.857). The median ALER was 0.102 (range: 0–0.325, n = 34) in the whole patient group and 0.102 (range: 0–0.325, n = 34) in the unmedicated patient group. The ALER values, similar to the NAT values, did not significantly differ (Mann–Whitney rank test U = 786, p = 0.969). In the retrieval part of the test phase, the median RER was 0.056 in the whole patient group (range: 0–0.333, n = 46) and 0.056 (range: 0–0.333, n = 34) in the unmedicated patient group, and this difference was not statistically significant (Mann–Whitney rank test U = 774, p = 0.937). In the generalization part of the test phase, the median GER was 0.125 in the whole patient group (range: 0–0.667, n = 46) and 0.083 (range: 0–0.667, n = 34) in the unmedicated patient group and this difference was not statistically significant (Mann–Whitney rank test U = 742, p = 0.698).

**Medicated versus unmedicated pediatric patients with Tourette syndrome.**   The performance of the medicated patient group did not differ significantly from the performance of the unmedicated patient group. The median NAT was 79.0 (range: 68–101, n = 12) for the medicated patient group and 78.5 (range: 42–202, n = 34) for the unmedicated patient group. There was no statistically significant difference in the NAT between these groups (Mann–Whitney rank test U = 185, p = 0.643). The median ALER was 0.106 (range: 0.056–0.250, n = 12) in the medicated patient group and 0.102 (range: 0–0.325, n = 34) in the unmedicated patient group. The ALER values, similar to the NAT values, did not significantly differ (Mann–Whitney rank test U = 208, p = 0.920). In the retrieval part of the test phase, the median RER was 0.083 in the medicated patient group (range: 0–0.139, n = 12) and 0.056 (range: 0–0.333, n = 34) in the unmedicated patient group, and this difference was not statistically significant (Mann–Whitney U test U = 196, p = 0.840). In the generalization part of the test phase, the median GER was 0.208 in the medicated patient group (range: 0–0.667, n = 12) and 0.083 (range: 0–0.667, n = 34) in the unmedicated patient group, and the difference was not statistically significant (Mann–Whitney rank test U = 164, p = 0.319).

## Comparison of the performances among the patients with TS, TS + ADHD, and TS + OCD/ASD

In the first step we have compared the performances in one multiple comparison of the three TS patient and the three control subgroups with Kruskal–Wallis ANOVA analysis. These

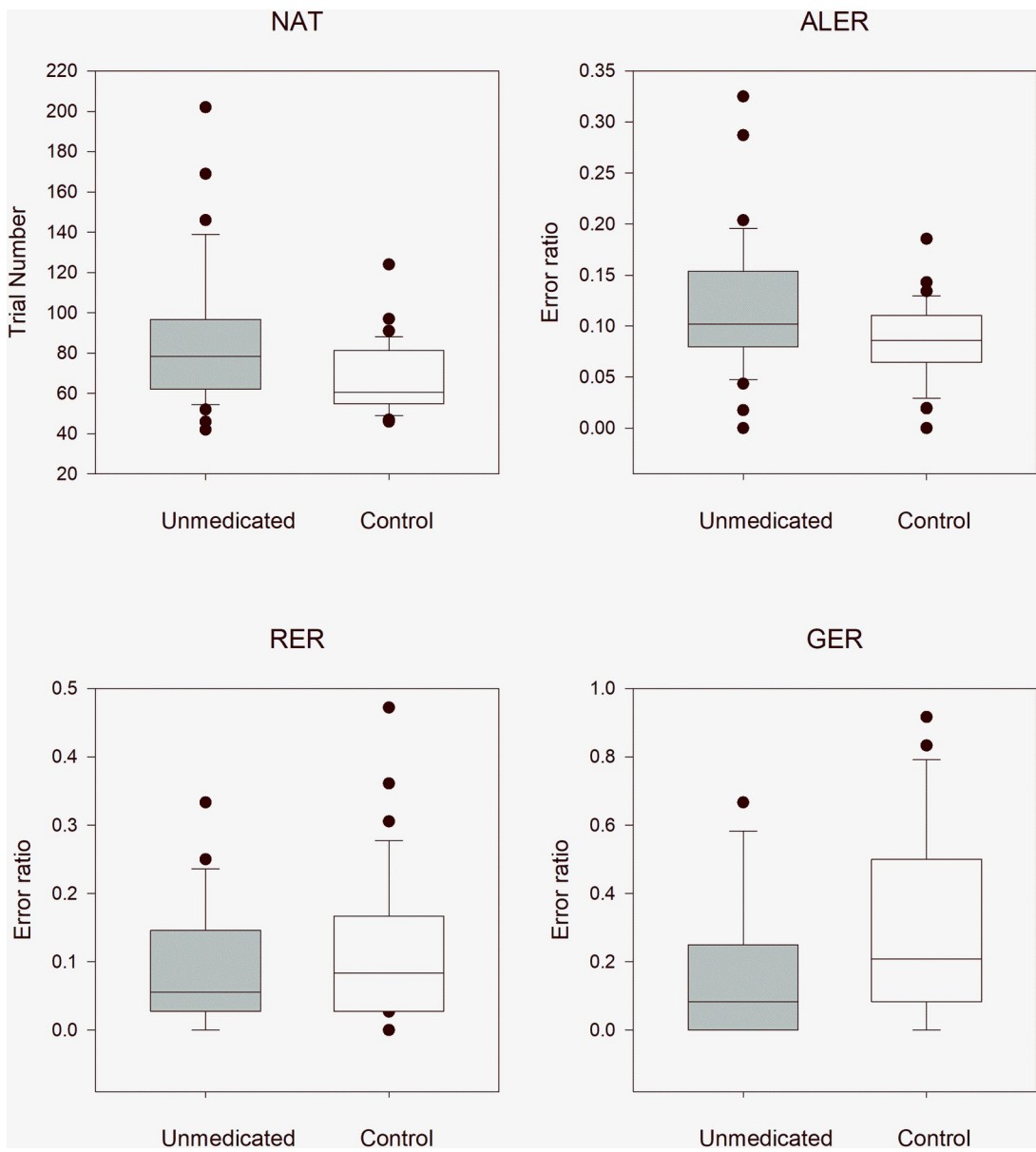

**Fig 3. Performance of the unmedicated pediatric patients with Tourette syndrome versus that of healthy control children in the visually guided equivalence learning paradigm.** NAT denotes the number of the necessary trials in the acquisition phase of the paradigm. ALER shows the error ratios in the acquisition phase of the paradigm. Lower diagrams denote the error ratios in the retrieval (RER) and generalization (GER) parts of the test phase, respectively. In each panel, the first column (gray) shows the performance of all unmedicated patients with Tourette syndrome, and the second column (white) denotes the performance of the control group. The lower margin of the boxes shows the 25th percentile; the line within the boxes marks the median; and the upper margin of the boxes indicates the 75th percentile. The error bars (whiskers) above and below the boxes indicate the 90th and 10th percentiles, respectively. The dots over and under the whiskers show the extreme outliers. The black stars indicate statistically significant differences ($p < 0.05$).

results revealed significant differences among the six subgroups in NAT ($\chi^2$ (5, N = 92) = 14.1829, p = 0.0145) and ALER ($\chi^2$ (5, N = 92) = 11.7513, p = 0.0384) but not in RER ($\chi^2$ (5, N = 92) = 1.9133, p = 0.861) and GER ($\chi^2$ (5, N = 92) = 3.3317, p = 0.6490). In the next step we have compared the performances among the three TS patient subgroups. There were no significant differences in the performances of the three patient groups (TS, TS + ADHD, and

**Table 2. The performances of the Tourette syndrome, Tourette syndrome and attention deficit hyperactivity disorder, and Tourette syndrome and obsessive compulsive disorder or autism spectrum disorder groups (with or without medication).**

|  | TS (n = 21) | TS + ADHD (n = 15) | TS + OCD/ASD (n = 10) | Kruskal–Wallis test |
|---|---|---|---|---|
| NAT | median: 79 range: 52–130 | median: 78 range: 42–202 | median:72 range: 46–169 | $\chi2(2, N = 46) = 0.0498, p = 0.975$ |
| ALER | median: 0.101 range: 0.018–0.325 | median: 0.120 range: 0–0.287 | median: 0.106 range:0.043–0.172 | $\chi2(2, N = 46) = 1.13, p = 0.568$ |
| RER | median: 0.056 range: 0–0.333 | median: 0.056 range: 0.028–0.222 | median: 0.097 range: 0.028–0.222 | $\chi2(2, N = 46) = 1.1, p = 0.577$ |
| GER | median: 0.083 range: 0–0.667 | median: 0.167 range: 0–0.667 | median: 0.208 range: 0–0.583 | $\chi2(2, N = 46) = 0.389 p = 0.823$ |

TS: Tourette syndrome, ADHD: attention deficit hyperactivity disorder, OCD/ASD: obsessive compulsive disorder or autism spectrum disorder, NAT: the number of the necessary trials in the acquisition phase of the paradigm, ALER: the error ratios in the acquisition phase of the paradigm, RER: the error ratios in the retrieval part of the test phase, and GER: the error ratio in the generalization part of the test phase.

TS + OCD/ASD with or without medication) for any of the investigated parameters. The results of the comparisons are shown in Table 2.

After the subtraction of the performances of the medicated patients from the analysis, there were no significant differences among the TS, TS + ADHD, and TS + OCD/ASD groups (Table 3).

To avoid the significant effect of the different performances of the three different control subgroups we have compared the performances of the control subgroups, too. We have found no significant differences in each of the investigated values among the control subgroups (Kruskal–Wallis ANOVA, NAT: $\chi^2$ (2, N = 46) = 3.7562, p = 0.153; ALER: $\chi^2$ (2, N = 46) = 3.5641, p = 0.168; RER: $\chi^2$ (2, N = 46) = 0.7136, p = 0.965; GER: $\chi^2$ (2, N = 46) = 0.16242, p = 0.922).

## Performance of the three TS groups versus their matched healthy control groups

To determine whether the significant findings described above originated in a similar manner for patients with Tourette syndrome without comorbidities and for patients with Tourette syndrome and comorbidities (TS + ADHD or TS + OCD/ASD) we separately compared the data of these three subpopulations with their matched healthy control groups.

**Children with Tourette syndrome without any comorbidities versus healthy control children.** We examined the difference between the performance of patients with Tourette syndrome and that of matched healthy controls. The median NAT was 79.0 (range: 52–130, n = 21) in the TS group and 60.0 (range: 46–124, n = 21) in the control group. The NAT was significantly higher in patients with Tourette syndrome (Mann–Whitney rank test U = 109, p = 0.005). The median ALER of patients with Tourette syndrome was 0.101 (range: 0.018–0.325, n = 21), and that of the healthy control group was 0.083 (range: 0–0.186, n = 21). The

**Table 3. The performances of the three unmedicated patient groups.**

|  | TS (n = 18) | TS + ADHD (n = 9) | TS + OCD/ASD (n = 7) | Kruskal–Wallis test |
|---|---|---|---|---|
| NAT | median: 79,5 range: 52–130 | median: 62 range: 42–202 | median: 89 range: 46–169 | $\chi2(2, N = 34) = 0.877, p = 0.645$ |
| ALER | median: 0.102 range: 0.018–0.325 | median: 0.097 range: 0–0.287 | median: 0.136 range: 0.043–0.172 | $\chi2(2, N = 34) = 0.694, p = 0.707$ |
| RER | median: 0.056 range: 0–0.333 | median: 0.056 range: 0.028–0.222 | median: 0.028 range: 0.028–0.222 | $\chi2(2, N = 34) = 0.535, p = 0.765$ |
| GER | median: 0.083 range: 0–0.667 | median: 0.083 range: 0–0.161 | median: 0.167 range: 0–0.583 | $\chi2(2, N = 34) = 0.255, p = 0.880$ |

TS: Tourette syndrome, ADHD: attention deficit hyperactivity disorder, OCD/ASD: obsessive compulsive disorder or autism spectrum disorder, NAT: the number of the necessary trials in the acquisition phase of the paradigm, ALER: the error ratios in the acquisition phase of the paradigm, RER: the error ratios in the retrieval part of the test phase, and GER: the error ratio in the generalization part of the test phase.

ALER values, similar to the NAT values, were higher in the TS group (Mann–Whitney rank test U = 142, p = 0.049). In the retrieval part of the test phase, the median RER in the TS group was 0.056, (range: 0–0.333, n = 21) and that in the matched healthy group was 0.083 (range: 0–0.361, n = 21). Although the RER was smaller in the Tourette syndrome group, this difference was not statistically significant (Mann–Whitney rank test U = 260, p = 0.327). In the generalization part of the test phase, the median GER was 0.083 (range: 0–0.667, n = 21) in the group of patients with Tourette syndrome and 0.167 (range: 0–0.917, n = 21) in the healthy control group. This difference was not statistically significant (Mann–Whitney rank test U = 270, p = 0.209, Fig 4).

**Patients with TS + ADHD versus healthy controls.** These comparisons revealed the same tendencies as those in the TS without any comorbidities and TS+OCD/ASD groups. The NAT and ALER values were higher in the TS + ADHD than in the control group, while the RER and GER values were lower in patients with TS+ADHD, but these differences did not significantly differ from those values of the matched control children. The median NAT was 78.0 (range: 42–202, n = 15) in the TS + ADHD group and 75.0 (range: 45–97, n = 15; Mann–Whitney rank test U = 89, p = 0.340) in the matched healthy control group. The median ALER was 0.120 (range: 0.0–0.287, n = 21) in the TS + ADHD group and 0.105 (range: 0.056–0.143, n = 15, Welch's test t(22,2) = -1.56, p = 0.138) in the control group. In the retrieval part of the test phase, the median RER was 0.056 (range: 0.028–0.222, n = 15) in the TS + ADHD group and 0.083 (range: 0.0–0.472, n = 15, Mann–Whitney rank test U = 120, p = 0.784) in the control group. In the generalization part of the test phase, the median GER was 0.167 (range: 0–0.667, n = 15) in the TS + ADHD group and 0.167 (range: 0–0.917, n = 15, Mann–Whitney rank test U = 140, p = 0.255) in the matched healthy control group (Fig 5).

**Patients with TS + OCD/ASD versus healthy controls.** This comparison revealed the same significant differences as were demonstrated above by the patients with Tourette syndrome without any comorbidities. The NAT and ALER values were significantly higher in the TS + OCD/ASD group than in the control group, while the RER and GER values did not differ between the TS + OCD/ASD and control groups. The median NAT was 72.0 (range: 46–169, n = 10) in the TS + OCD/ASD group and 60.5 (range: 49–84, n = 10; independent samples t-test t(18) = -2.21, p = 0.041) in the matched healthy control group. The median ALER was 0.106 (range: 0.043–0.172, n = 10) in the TS + OCD/ASD group and 0.083 (range: 0.019–0.125, n = 10, independent samples t-test t(18) = -2.48, p = 0.023) in the control group. In the retrieval part of the test phase, the median RER was 0.097 (range: 0.028–0.222, n = 10) in the TS + OCD/ASD group and 0.083 (range: 0.027–0.250, n = 10, independent samples t-test t(18) = 0.335, p = 0.741) in the control group. In the generalization part of the test phase, the median GER was 0.208 (range: 0–0.583, n = 10) in the TS + OCD/ASD group and 0.208 (range: 0–0.750, n = 10, Mann–Whitney rank test U = 51.5, p = 0.939) in the matched healthy control group (Fig 6).

## Discussion

The Rutgers Acquired Equivalence Test (face and fish test, [35]), which investigates visually guided associative learning in humans, has a well-defined neurological background. The acquisition phase, which primarily depends on the function of the basal ganglia [35, 50] tests the association between two different visual stimuli. The test phase, in which the previously learned associations (retrieval part) and new, but acquisition-based, predictable associations (generalization part) are evaluated, primarily depends on the hippocampi and the mediotemporal lobe [35, 50]. These cognitive functions were previously investigated in adult neurological and psychiatric patients which were shown to be related to dysfunction of the basal ganglia

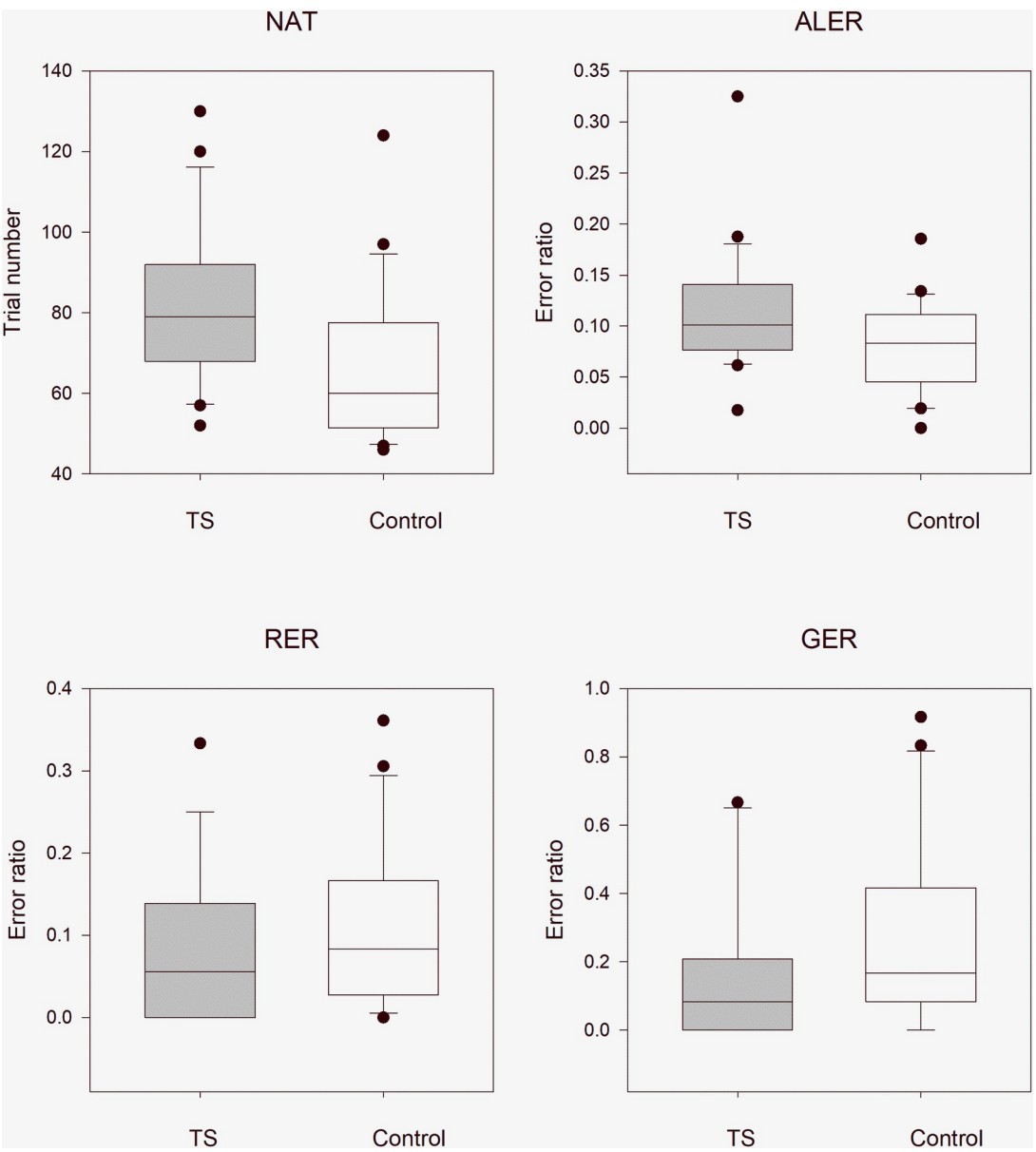

**Fig 4. Performance of the patients with Tourette syndrome without comorbidities versus that of matched healthy control children in the visually guided equivalence learning paradigm.** NAT denotes the number of the necessary trials in the acquisition phase of the paradigm. ALER shows the error ratios in the acquisition phase of the paradigm. Lower diagrams denote the error ratios in the retrieval (RER) and generalization (GER) parts of the test phase, respectively. In each panel, the first column (gray) shows the performance of the patients with Tourette syndrome without comorbidities, and the second column (white) denotes the performance of the control group. The lower margin of the boxes shows the 25th percentile; the line within the boxes marks the median; and the upper margin of the boxes indicates the 75th percentile. The error bars (whiskers) above and below the boxes indicate the 90th and 10th percentiles, respectively. The dots over and under the whiskers show the extreme outliers. The black stars indicate statistically significant differences (p < 0.05).

and the hippocampi (i.e., Parkinson's disease [34, 35], Alzheimer's disease [51], and migraine without aura [53]). However, the present study is the first to describe the alteration of visual associative learning in a group of children with Tourette syndrome with and without comorbidities. This finding is interesting because only in rare cases have significant impairments in

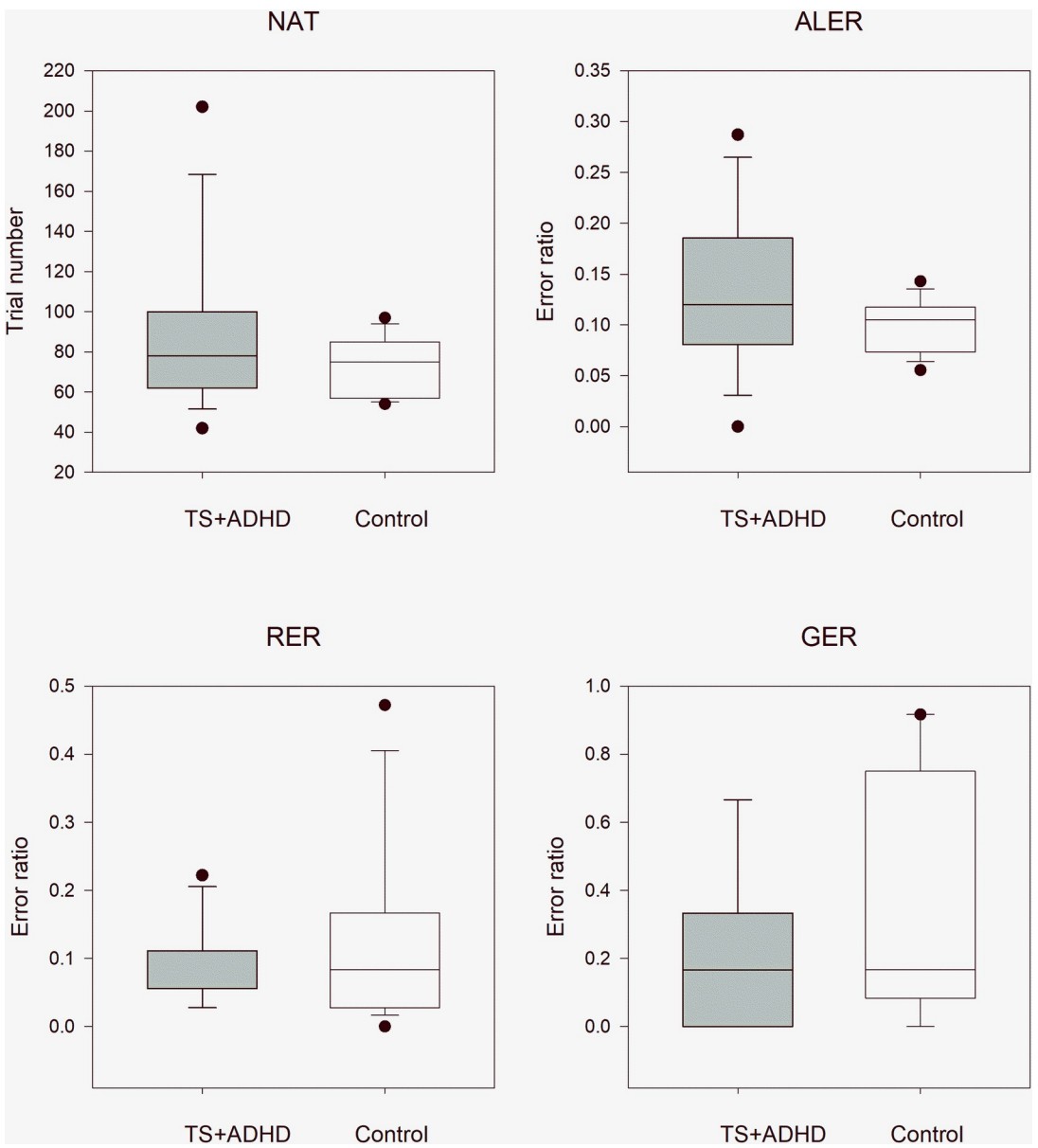

**Fig 5. Performance of patients with concomitant Tourette syndrome and attention deficit hyperactivity disorder patients versus that in matched healthy control children in the visually guided equivalence learning paradigm.** NAT denotes the number of the necessary trials in the acquisition phase of the paradigm. ALER shows the error ratios in the acquisition phase of the paradigm. Lower diagrams denote the error ratios in the retrieval (RER) and generalization (GER) parts of the test phase, respectively. In each panel, the first column (gray) shows the performance of the patients with Tourette syndrome (TS) and attention deficit hyperactivity disorder (ADHD), and the second column (white) denotes the performance of the control group. The lower margin of the boxes shows the 25th percentile; the line within the boxes marks the median; and the upper margin of the boxes indicates the 75th percentile. The error bars (whiskers) above and below the boxes indicate the 90th and 10th percentiles, respectively. The dots over and under the whiskers show the extreme outliers. The black stars indicate statistically significant differences (p < 0.05).

any cognitive functions been described in Tourette syndrome. Tourette syndrome is strongly related to dysfunction of the basal ganglia and the frontal associative cortex. Because of the involvement of the basal ganglia in the pathogenesis of Tourette syndrome, the acquisition phase, which mainly depends on the basal ganglia, was primarily affected in the associative

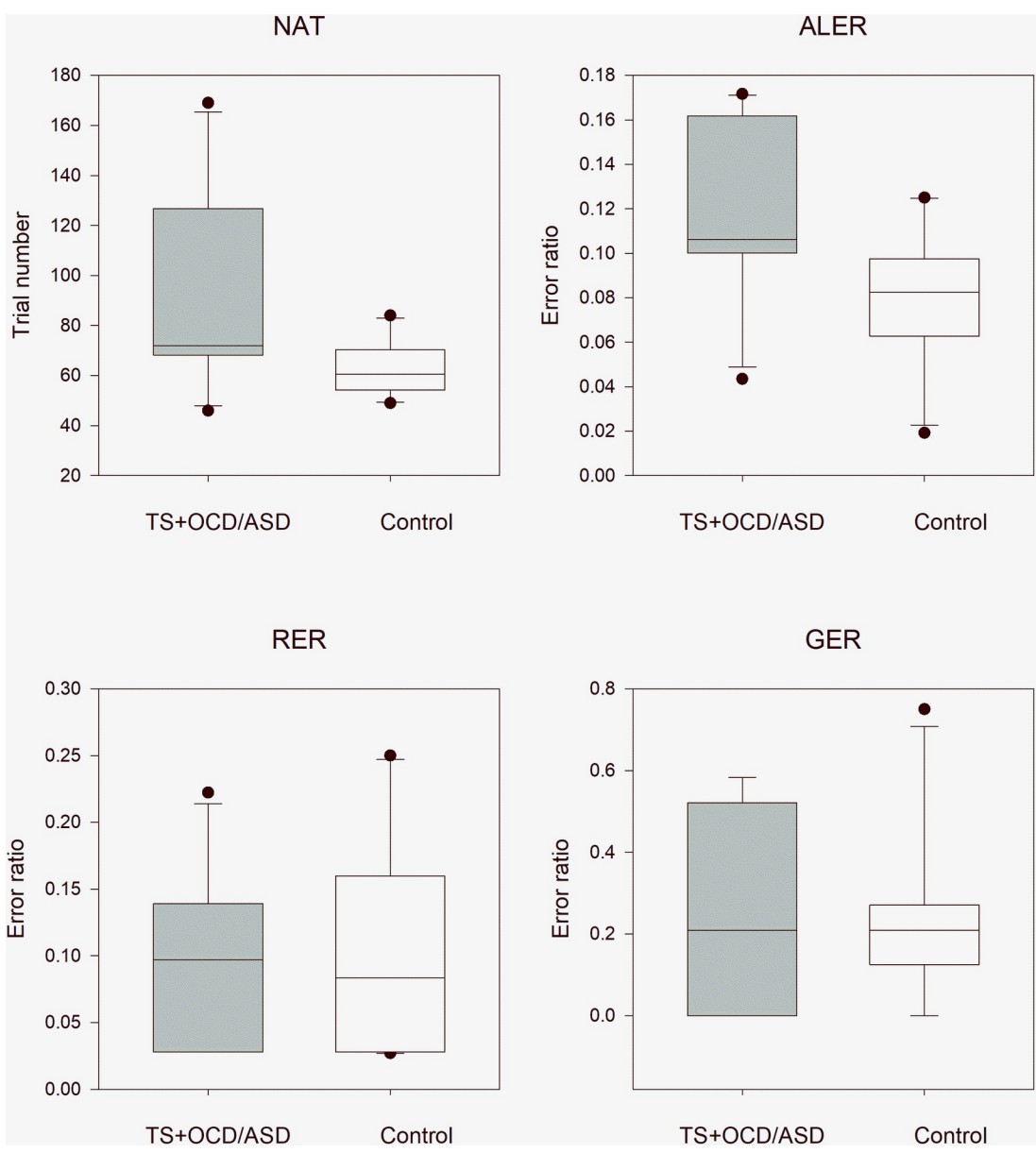

**Fig 6. Performance of patients with concomitant Tourette syndrome and obsessive compulsive disorder or autism spectrum disorder versus that of matched healthy control children in the visually guided equivalence learning paradigm.** NAT denotes the number of the necessary trials in the acquisition phase of the paradigm. ALER shows the error ratios in the acquisition phase of the paradigm. Lower diagrams denote the error ratios in the retrieval (RER) and generalization (GER) parts of the test phase, respectively. In each panel, the first column (gray) shows the performance of the patients with Tourette syndrome (TS) and obsessive compulsive disorder (OCD) or autism spectrum disorder (ASD), and the second column (white) denotes the performance of the control group. The lower margin of the boxes shows the 25th percentile; the line within the boxes marks the median; and the upper margin of the boxes indicates the 75th percentile. The error bars (whiskers) above and below the boxes indicate the 90th and 10th percentiles, respectively. The dots over and under the whiskers show the extreme outliers. The black stars indicate statistically significant differences ($p < 0.05$).

learning test. Based on our results, all patients with Tourette syndrome made the associations with less effectiveness than healthy control children. However, the retrieval and generalization parts of the test phase, which primarily depend on the function of the hippocampi, were not negatively affected by Tourette syndrome. Because of the compensation of the weaker

acquisition building, even better performances were found in these phases of the paradigm, although these differences were not statistically significant [36, 47].

Our results demonstrated that in the acquisition phase, the performance (NAT and ALER) of all patients (TS, TS + ADHD, and TS + OCD/ASD) was significantly weaker than in the sex, age, and IQ level-matched healthy control group. The question arises whether the alterations in equivalence learning in all patients with Tourette syndrome were primarily due to TS or its most common comorbidity, ADHD. In most cases, Tourette syndrome and ADHD, which seems to plays a major role, are jointly responsible for the alterations in cognitive functions [44, 59–61]. We compared the performances of the three patient groups and found no significant difference among their performances. This finding does not support the predominant role of ADHD in the described alterations in the acquisition phase of the associative learning task. The comparison of each patient group with its matched healthy control group revealed significantly increased NAT or ALER values in patients with Tourette syndrome without any comorbidities and TS + OCD/ASD but not in patients with TS + ADHD. These results together could suggest that concomitant ADHD and TS was not primarily responsible for the visual acquisition learning deficits in patients with Tourette syndrome. This is in contrast with previous findings that ADHD is primarily responsible for the alteration of cognitive functions in patients with TS + ADHD [33, 37, 38, 43]. Therefore, the visually guided acquired equivalence learning, similar to stimulus-response or habit learning [24, 45], which is mediated by the dorsal frontostriatal pathways, is more attributable to Tourette syndrome than ADHD, despite ADHD symptoms affecting the dorsolateral frontostriatal circuits [62].

The volume of the hippocampi is significantly larger in patients with pure Tourette syndrome than that of their healthy counterparts [63], and no explicit memory (which is connected to the hippocampus) deficits were reported in children with Tourette syndrome [45, 47]. Our results are in line with these findings. The performance in the retrieval and generalization parts of the test phase, which are primarily related to the hippocampi was not worse in the entire group of patients with Tourette syndrome with and without comorbidities. Concerning the three investigated subpopulations of the patients with Tourette syndrome (TS without comorbidities, TS + ADHD, and TS + OCD/ASD), the RER and GER values did not differ from those of the matched healthy control children.

Another question is the possible influence of medication on the performance of patients with Tourette syndrome with or without comorbidities. Because of the relatively low number of cases in the comorbid groups, we could not perform a valid comparison between the performance of medicated and unmedicated TS + ADHD and TS + OCD/ASD patients. Thus, we used the entire TS population (TS without comorbidities, TS + ADHD, and TS + OCD/ASD) to get information about the possible role of medication. The performances in the acquisition phase of the associative learning task in unmedicated TS pediatric patients, similar to the entire TS population, were significantly weaker than those of the matched healthy control children. The comparison of the performances of the entire and the unmedicated TS patient groups revealed no differences. Similarly, we found no differences between the performances of the entire population of medicated and unmedicated pediatric patients with Tourette syndrome. These findings collectively suggest that medication had no or only a weak influence on our results.

In this study, we functionally confirmed the results of neuroimaging [14–16, 23] and functional studies that the dorsal frontostriatal circuits are strongly affected in Tourette syndrome, and these circuits are critical to the acquisition process of visually guided associative learning [46, 47]. The hippocampus mediated recall of previously learned associations, and the building of new but acquisition-based, predictable associations were not altered in Tourette syndrome.

## Supporting information

**S1 Table. The data of the psychophysical performances of 46 TS patients and the 46 matched controls, which were used in the present study.** TS: Tourette syndrome, ADHD: attention deficit hyperactivity disorder, OCD/ASD: obsessive compulsive disorder or autism spectrum disorder, NAT: the number of the necessary trials in the acquisition phase of the paradigm, ALER: the error ratios in the acquisition phase of the paradigm, RER: the error ratios in the retrieval part of the test phase, and GER: the error ratio in the generalization part of the test phase.
(XLSX)

## Acknowledgments

The authors thank Emese Bognár, Kristóf Kollár, András Puszta, Xenia Katona, Nóra Cserháti, Nándor Görög and Dóra Dózsai for their help in conducting the investigation and data collection and all the participants for engaging in the research. The open access publication was founded by the University of Szeged Open Access Fund, Grant number: 4780.

## Author Contributions

**Conceptualization:** Gabriella Eördegh, Péter Nagy, Attila Őze, Attila Nagy.

**Data curation:** Gabriella Eördegh, Ákos Pertich, Zsanett Tárnok, Zsófia Giricz, Orsolya Hegedűs, Dóra Merkl, Diána Nyujtó, Szabina Oláh, Attila Őze, Réka Vidomusz.

**Formal analysis:** Gabriella Eördegh.

**Funding acquisition:** Attila Nagy.

**Investigation:** Gabriella Eördegh, Ákos Pertich, Zsanett Tárnok, Péter Nagy, Balázs Bodosi, Zsófia Giricz, Orsolya Hegedűs, Dóra Merkl, Diána Nyujtó, Szabina Oláh, Attila Őze, Réka Vidomusz.

**Methodology:** Gabriella Eördegh, Balázs Bodosi, Attila Őze, Attila Nagy.

**Project administration:** Gabriella Eördegh, Zsanett Tárnok.

**Resources:** Zsanett Tárnok, Péter Nagy, Attila Nagy.

**Software:** Balázs Bodosi, Attila Őze.

**Supervision:** Attila Nagy.

**Validation:** Gabriella Eördegh, Attila Őze.

**Visualization:** Ákos Pertich, Zsófia Giricz.

**Writing – original draft:** Gabriella Eördegh.

**Writing – review & editing:** Ákos Pertich, Zsófia Giricz, Attila Nagy.

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
