## [Decision Letter · Decision Letter 0]

17 Feb 2020

PONE-D-19-30158

Impairment of visually guided associative learning in Tourette syndrome with and without ADHD

PLOS ONE

Dear Dr Eördegh,

Thank you for submitting your manuscript to PLOS ONE. After careful consideration, we feel that it has merit but does not fully meet PLOS ONE’s publication criteria as it currently stands. Therefore, we invite you to submit a revised version of the manuscript that addresses the points raised during the review process.

Please re-analyze the data incorporating the possible effects of medication into the analyzes. Please have the manuscript examined by a professional language editor.

We would appreciate receiving your revised manuscript by Apr 02 2020 11:59PM. To enhance the reproducibility of your results, we recommend that if applicable you deposit your laboratory protocols in protocols.io, where a protocol can be assigned its own identifier (DOI) such that it can be cited independently in the future. For instructions see: http://journals.plos.org/plosone/s/submission-guidelines#loc-laboratory-protocols

We look forward to receiving your revised manuscript.

Kind regards,

Alexandra Kavushansky, PhD

Academic Editor

PLOS ONE

Journal Requirements:

Reviewers' comments:

Reviewer's Responses to Questions

**Comments to the Author**

1. Is the manuscript technically sound, and do the data support the conclusions?

Reviewer #1: No

Reviewer #2: Partly

2. Has the statistical analysis been performed appropriately and rigorously? 

Reviewer #1: No

Reviewer #2: No

3. Have the authors made all data underlying the findings in their manuscript fully available?

Reviewer #1: No

Reviewer #2: No

4. Is the manuscript presented in an intelligible fashion and written in standard English?

Reviewer #1: No

Reviewer #2: No

5. Review Comments to the Author

Reviewer #1: This manuscript reports on original data determining the associative learning abilities in patients with Tourette syndrome with and without ADHD. This is informative to the field, but as written, the data do not support the conclusions and the statistical analyses are lacking in part (see Major Comments 3 and 7). Some ambiguity exists in description of the patient population and the study design (see Major Comments 1, 3-7).

Major Comments

1. A frequent inquiry in the literature is how behavior in Tourette syndrome differs by the presence of comorbidities, especially ADHD. Though the title indicates this as a consideration in this manuscript, ADHD is not discussed adequately as a factor in the Introduction. For example, the introduction on cognitive function does not indicate which references controlled for the presence of comorbid ADHD. Thus, the reader assumes any deficiencies listed are due to pure Tourette Syndrome, whereas some groups cited find them due to comorbid ADHD. This understanding of comorbid ADHD is expressed too late in line 80. Rather than introducing neuropsychological functions first and the impact of ADHD second, consider doing so in parallel.

2. Many of the studies cited, especially on learning disabilities and executive functions, are from work prior to 2000. Recent studies should also be considered, as these more often include consideration of comorbid ADHD. (e.g., Jeter et al., 2015; Termine et al., 2016; Openneer et al., 2019).

3. Please describe the patient sample more thoroughly. For example, the authors state patients with Tourette syndrome are usually of normal intelligence, yet those included in this cohort score a 2 (very high) on the intelligence scale used. Also, what is the average tic severity of the cohorts at the time of testing? The waxing and waning of symptoms may be an indicator of fronto-striato-cortical function, and perhaps cognitive function. Tic severity has been a considered factor in a variety of abilities in Tourette Syndrome (e.g., Burd et al., 2005; Jeter et al. 2015; Yaniv et al., 2018). Finally, what medications are the patients taking?

4. It is not clear why the many neuropsychological functions are discussed in the Introduction. How, if at all, does associative learning relate to the many other neuropsychological functions listed? The underlying physiology of the task is explained and linked to a prediction of performance on the task. If, though, associative learning is related to one or more functions, how should we anticipate patients with Tourette syndrome to perform on this task of associative learning?

5. The Methods report that 47 children with Tourette syndrome participated, yet data are presented for only 36. If data from the 11 children with Tourette syndrome and OCD are not included in this manuscript, do not mention them. If these children did complete the task, including their data in this manuscript produces a more complete understanding of associative learning in Tourette syndrome.

6. Description of the task must be clearer. Divide description of the task into multiple paragraphs. Please revise the figure to include the colored fish not associated/selected. For example, show that faces A1 and A2 are first associated with X1 (yellow fish), and that it is Y1 (green fish) that is the wrong answer. How did the participant know when to abandon the first associations (with A1, A2) and develop new (with A1)? What are the correct answers in the generalization phase? The text explains that having learned A1 and A2 are equivalent; the participant will associate A2 with X2. Show in the figure that the alternative wrong fish is Y2 (blue).

7. The authors must consider the impact of medications on patient performance. Many studies show the importance of controlling for this factor, preferably in the study design, but post hoc at a minimum. Contrary to most articles on neuropsychological functions, this study found that Tourette syndrome, not comorbid ADHD, contributed to deficits in associative learning. Yet, more patients with Tourette syndrome and ADHD were on medications (6/15=40%) than Tourette syndrome alone (3/21=14%), possibly masking the ill effects of ADHD on associative learning. Add analysis of patient performance by subtype, but only include those off medications. Discuss the findings in the Results and limitations in the Discussion.

8. If associative learning deficits truly are attributed to Tourette Syndrome, and not comorbid conditions, discuss why this may be, particularly since it is the comorbid ADHD that is responsible for deficits in other neuropsychological functions. Link this to possible physiology.

Minor Comments

1. The Abstract (and Figure 1) describes the task as three main parts: acquisition, retrieval and generalization phases. Yet, the Introduction and Methods describe the task as having two parts: acquisition and test, with the latter having retrieval and generalization parts. Reconciling this language is important to conceptualizing the task.

2. The Introduction describes Tourette syndrome as frequent, but most readers will not consider 1% prevalence as frequent.

3. Please use the phrase, “patients with Tourette syndrome” rather than “Tourette Syndrome patients.” Some find the latter offensive, as it uses a disease as the descriptor of the person (i.e., the person is inseparable from the disease), rather than describing a person with a disease (i.e., the person has an identity apart from the disease). So, too, the phrase “mentally retarded” is now replaced with “intellectually disabled.”

4. The Introduction states “only in rare cases…” Please be more specific, as this can be interpreted as “the rare child” or “the rare cognitive function.”

5. The relevance of associative learning to daily life is not explained. Even simple statements about how a deficit in this domain can affect academic performance could help.

6. Insert Table 1 after the explanation of how IQ is estimated.

7. Define ADHD in Tables and captions.

8. In defining NAT, RER, ALER, and GER, designate which are numerators, denominators in these ratios.

9. On line 374, the hippocampi of patients with Tourette syndrome were larger than whose? TS+ADHD or controls?

10. References 3 and 9 are the same article.

Reviewer #2: The manuscript reports the results of a study comparing the associative learning performance of children with Tourette syndrome (TS) in the absence of any comorbidity (N=21) with children with TS and ADHD (N=15), on associative learning in a variant of the Rutgers Acquired Equivalence Test (adapted for use in Hungarian children). N=36 matched controls were also tested. Children with TS comorbid with ASD or OCD (N=11) are mentioned but I can’t see that these data have been included. These could well be publishable findings but I’m struggling to follow the conclusions in the present version of the manuscript.

Points to be addressed

As is to be expected for this kind of work, the sample sizes are reasonable, but likely insufficient to take any differences attributable to medication into account. N=3 TS and N=6 TS+ADHD children were medicated and medication has previously been found to affect associative learning in both TS and ADHD. I’m not sure that the medication details have been provided and I can’t see that potentially confounded effects of medication have been discussed at all within the present manuscript.

If the lack of difference in performance between TS and TS+ADHD groups suggests that the effects shown in Figure 2 (differences in NAT and ALER) are TS-mediated, should the same profile not also be shown in the 3rd TS group (comorbid with ASD or OCD)?

The third TS group don’t seem to be shown in the figures, nor are the data considered with and without the exclusion of participants on medication?

Ln 123-128 - I didn’t understand the rationale to categorise IQ scores: like age, IQ is a continuous variable and to categorise the participants’ scores into IQ ranges may result in a loss of statistical power. Please explain. It’s also not clear how we are to understand the categories which run from ‘extremely high’ through to ‘very low’ with no further definition of these categories.

Ln 361-365 – I can’t see how this conclusion about the compensatory effect of ADHD follows. I thought the TS and TS+ADHD groups were not significantly different from each other? Now we’re getting a different comparison (each of the sub-groups with the controls) and the TS-ADHD group also happens to be smaller (N=15). Plus medication does not seem to have been taken into account.

The writing is mostly understandable but the manuscript would benefit from further editing by a fluent English speaker, this is not really the job of the reviewer. For example, the Abstract which is most visible (and all some will read) could be better written (some specific suggestions below).

Ln 21-22 - The first sentence is a little awkward.

Ln22-23 – ‘the majority of the cognitive functions’ - ‘the majority of cognitive functions’

Ln 23 - ‘only little evidence’ – suggest rephrase

Ln 26 – ‘The acquired equivalence learning…’ - ‘Acquired equivalence learning…’

Ln 32 – ‘the entire patient group’ – please be more specific, presumably TS+ADHD

Ln 33 – ‘associations with lower effectiveness’ – please rephrase

Ln 36-37 – ‘parts of the test phase’ – plural so ‘depend…’ (and sentence seems to be missing a comma after hippocampus)

Minor comments and typos

Ln 140 - iOS – should be defined?

Ln 184 – misplaced (and uninformative) figure caption text.

Ln 211-212 – Data availability statement and link will need updating. I’m not sure I see the point of providing the data for peer review only at the first revision if the data have not been provided at this point (for peer review prior to the point of being on a ‘revise and resubmit’ ticket)? Elsewhere (in the additional information boxes) it says all relevant data are within the manuscript and its Supporting Information files but I can’t see that the data has in fact been provided at this point. The Supporting Information files seem to be figures.

Ln 238-248 – I think this para is the figure legend.

Ln 272-277 – I think this para is the figure legend.

Ln 297-302 – I think this para is the figure legend (ditto for other figures, why is the legend in the text?)

Ln 579 – please make the figure caption self-explanatory rather than referring the reader back to the text.

Ln 583 - are the data for TS and TS+ADHD participants in fact shown separately in Figure 2? (As implied by the figure header.) If we’re looking at all the TS data relative to matched controls why not also include TS+OCD/ASD?

Ln 608 - ‘syndrom’ (typo)

Ln 612 - please make the figure caption self-explanatory rather than referring the reader back to an earlier figure legend.

6. PLOS authors have the option to publish the peer review history of their article (what does this mean?). If published, this will include your full peer review and any attached files.

Reviewer #1: Yes: Cameron B. Jeter

Reviewer #2: No

---

## [Author Response · Author response to Decision Letter 0]

17 Apr 2020

First of all, we would like to express our gratitude for the scholarly and highly helpful criticism of both Referees, which helped us to improve the quality of our study. We accepted all suggestions of both Referees and made changes accordingly. The suggestions are answered below itemized; textual changes are marked in the final manuscript.

Major Comments

1. A frequent inquiry in the literature is how behavior in Tourette syndrome differs by the presence of comorbidities, especially ADHD. Though the title indicates this as a consideration in this manuscript, ADHD is not discussed adequately as a factor in the Introduction. For example, the introduction on cognitive function does not indicate which references controlled for the presence of comorbid ADHD. Thus, the reader assumes any deficiencies listed are due to pure Tourette Syndrome, whereas some groups cited find them due to comorbid ADHD. This understanding of comorbid ADHD is expressed too late in line 80. Rather than introducing neuropsychological functions first and the impact of ADHD second, consider doing so in parallel.

Response: According to the suggestion of the reviewer we have restructured the Introduction part of the manuscript. 

2. Many of the studies cited, especially on learning disabilities and executive functions, are from work prior to 2000. Recent studies should also be considered, as these more often include consideration of comorbid ADHD. (e.g., Jeter et al., 2015; Termine et al., 2016; Openneer et al., 2019).

Response: According to the suggestion we have removed the not obligatory references before 2000, and added the recommended new ones.

3. Please describe the patient sample more thoroughly. For example, the authors state patients with Tourette syndrome are usually of normal intelligence, yet those included in this cohort score a 2 (very high) on the intelligence scale used. Also, what is the average tic severity of the cohorts at the time of testing? The waxing and waning of symptoms may be an indicator of fronto-striato-cortical function, and perhaps cognitive function. Tic severity has been a considered factor in a variety of abilities in Tourette Syndrome (e.g., Burd et al., 2005; Jeter et al. 2015; Yaniv et al., 2018). Finally, what medications are the patients taking?

Response: We have deleted the IQ part of the Table 1. because of its low relevance to our study. Additionally, we have deleted the criticized sentence from the Introduction part of the manuscript. The difference from average IQ level in our cohort could arise from the higher educated parents, who have recognized the problem and bring their children to this special health care system. Another explanation could be that the Raven matrices are not as sensitive as the Wechsler intelligence test.

We added the still missing tic severity data of the cohort.

We described in detail the medication of the patient groups in Subjects subsection of methods.

4. It is not clear why the many neuropsychological functions are discussed in the Introduction. How, if at all, does associative learning relate to the many other neuropsychological functions listed? The underlying physiology of the task is explained and linked to a prediction of performance on the task. If, though, associative learning is related to one or more functions, how should we anticipate patients with Tourette syndrome to perform on this task of associative learning?

Response: We reduced the previous mentioned but to this manuscript not obligatory connected neuropsychological functions from the Introduction. We have described how the acquired equivalence learning can be connected to the above discussed neuropsychological functions. We have chosen a simple test (which could be done with success by the intellectually disabled individuals, too (de Rose JC, McIlvane WJ, Dube WV, Stoddard LT. Stimulus class formation and functional equivalence in moderately retarded individuals' conditional discrimination. Behav Processes. 1988;17(2):167-75.), which could be performed most probably with success by the Tourette syndrome patients, too. Our question was the effectivity in the applied learning paradigm.

5. The Methods report that 47 children with Tourette syndrome participated, yet data are presented for only 36. If data from the 11 children with Tourette syndrome and OCD are not included in this manuscript, do not mention them. If these children did complete the task, including their data in this manuscript produces a more complete understanding of associative learning in Tourette syndrome.

Response: In the Tourette syndrome and other (OCD and autism spectrum disorder, referred as to Tourette syndrome and OCD/ASD group) group ten of eleven children were able to complete the test (7 OCD and 3 ASD comorbid patient). We have involved this new group in the analysis. See new results of the manuscript). Thus the number of the participants increased to 46 in both patient and control groups, respectively. We also discussed these new results.

6. Description of the task must be clearer. Divide description of the task into multiple paragraphs. Please revise the figure to include the colored fish not associated/selected. For example, show that faces A1 and A2 are first associated with X1 (yellow fish), and that it is Y1 (green fish) that is the wrong answer. How did the participant know when to abandon the first associations (with A1, A2) and develop new (with A1)? What are the correct answers in the generalization phase? The text explains that having learned A1 and A2 are equivalent; the participant will associate A2 with X2. Show in the figure that the alternative wrong fish is Y2 (blue).

Response: In order to make the description of the paradigm more accurate we have introduced the suggested changes to the manuscript. Accordingly, we have prepared a new figure about the paradigm (see new Figure 1)

7. The authors must consider the impact of medications on patient performance. Many studies show the importance of controlling for this factor, preferably in the study design, but post hoc at a minimum. Contrary to most articles on neuropsychological functions, this study found that Tourette syndrome, not comorbid ADHD, contributed to deficits in associative learning. Yet, more patients with Tourette syndrome and ADHD were on medications (6/15=40%) than Tourette syndrome alone (3/21=14%), possibly masking the ill effects of ADHD on associative learning. Add analysis of patient performance by subtype, but only include those off medications. Discuss the findings in the Results and limitations in the Discussion.

Response: We analyzed the impact of medication on the performance in a new subsection in Results from three different aspects. These result support that the medication has no significant modulatory effects on the results. Thus the separate comparison of the performances of the entire three patient groups (TS, TS+ADHD, TS+OCD/ASD) with their matched control group we provided in the revised manuscript.

The case number of the unmedicated TS+ADHD and TS+OCD/ASD groups would be under 10 and this calls into question the relevance of statistical analysis. However, we have performed the comparison of the unmedicated groups with their matched controls and we provide the results here in these tables. These results are basically same to the results of the entire (medicated and unmedicated together) patient subgroups. Thus we have decided to publish the results of the entire TS populations.

 TS (n=18) TS control (n=18) Mann-Whitney rank test

NAT median: 79.5

range: 52-130 median: 60

range: 46-124 U=246,

p = 0.009

ALER median: 0.102

rage: 0.018-0.325 median: 0.083

range: 0-0.186 U=221,

p = 0.064

RER median: 0.056

range: 0-0.333 median: 0.083

range: 0-0.361 U=140,

p = 0.482

GER median: 0.083

range: 0-0.667 median: 0.167

range: 0-0.917 U=126

p = 0.252

Table 1 Performance of the unmedicated patient with Tourette syndrome group compared with matched healthy control group

TS means Tourette syndrome. NAT means the number of the necessary trials in the acquisition phase of the paradigm, ALER means the error ratios in the acquisition phase of the paradigm, RER means the error ratios in the retrieval and GER in the generalization parts of the test phase.

 TS+ADHD (n=9) TS+ADHD control (n=9) Mann-Whitney rank test

NAT median: 62

range: 42-202 median: 75

range: 56-97 U=47.5,

p = 0.565

ALER median: 0.097

rage: 0-0.287 median: 0.105

range: 0.07-0.143 U=737,

p = 0.480

RER median: 0.056

range: 0.028-0.222 median: 0.083

range: 0.028-0.472 U=35.5,

p = 0.687

GER median: 0.083

range: 0-0.5 median: 0.167

range: 0-0.917 U=28,

p = 0.280

Table 2 Performance of the unmedicated patient with Tourette syndrome + ADHD group compared with matched healthy control group

TS means Tourette syndrome, ADHD means Attention Deficit Hyperactivity Disorder. NAT means the number of the necessary trials in the acquisition phase of the paradigm, ALER means the error ratios in the acquisition phase of the paradigm, RER means the error ratios in the retrieval and GER in the generalization parts of the test phase.

 TS+OCD/ASD (n=7) TS+OCD/ASD control (n=7) Statistical test 

NAT median: 89

range: 46-169 median: 61

range: 49-84 Welch test t(9.22)=2.08,

p = 0.076

ALER median: 0.136

rage: 0.043-0.172 median: 0.072

range: 0.019-0.125 t test t(12)=2.16,

p = 0.052

RER median: 0.028

range: 0.028-0.222 median: 0.111

range: 0.028-0.250 Mann-Whitney test U=20.5,

p = 0.645

GER median: 0.167

range: 0-0.583 median: 0.250

range: 0-0.750 t test t(12)=-0.822,

p = 0.427

Table 3 Performance of the unmedicated patient with Tourette syndrome + OCD/ASD group compared with matched healthy control group

TS means Tourette syndrome, OCD/ASD means the third patient group with OCD and ASD comorbidity. NAT means the number of the necessary trials in the acquisition phase of the paradigm, ALER means the error ratios in the acquisition phase of the paradigm, RER means the error ratios in the retrieval and GER in the generalization parts of the test phase.

8. If associative learning deficits truly are attributed to Tourette Syndrome, and not comorbid conditions, discuss why this may be, particularly since it is the comorbid ADHD that is responsible for deficits in other neuropsychological functions. Link this to possible physiology.

Response: We have deleted the speculations about the primary effect of the TS or the ADHD in the alterations of learning functions of the patients. We have modified the abstract accordingly. We have described in the discussion only, which seems to be clear based on our results that not the ADHD is alone/primarily responsible for the associative learning disabilities of the entire TS patient group.

Minor Comments

1. The Abstract (and Figure 1) describes the task as three main parts: acquisition, retrieval and generalization phases. Yet, the Introduction and Methods describe the task as having two parts: acquisition and test, with the latter having retrieval and generalization parts. Reconciling this language is important to conceptualizing the task.

Response: We have made clearer the descriptions of the different parts of the learning task in each chapter of the manuscript.

2. The Introduction describes Tourette syndrome as frequent, but most readers will not consider 1% prevalence as frequent.

Response: We have deleted the word “frequent”.

3. Please use the phrase, “patients with Tourette syndrome” rather than “Tourette Syndrome patients.” Some find the latter offensive, as it uses a disease as the descriptor of the person (i.e., the person is inseparable from the disease), rather than describing a person with a disease (i.e., the person has an identity apart from the disease). So, too, the phrase “mentally retarded” is now replaced with “intellectually disabled.”

Response: We have changed the text according the suggestion of the reviewer.

4. The Introduction states “only in rare cases…” Please be more specific, as this can be interpreted as “the rare child” or “the rare cognitive function.”

Response: Done.

5. The relevance of associative learning to daily life is not explained. Even simple statements about how a deficit in this domain can affect academic performance could help.

Response: We added to the Introduction a typical application of the associative learning in the daily life.

6. Insert Table 1 after the explanation of how IQ is estimated.

Response: In the study, we used the Standard Progressive Matrices (SPM) at volunteers over 12 years of age and the Colored Progressive Matrices (CPM) below the age of 12 years. The Hungarian standard values of the two tests cannot be completely matched. The SPM values are in age-independent categories (7 levels), while the CPM values are in the Hungarian standard for age-dependent categories (8 levels, over 95, 90-95, 75-90, 50-75, 25-50, 10-25, 5-10, and bellow 5 centiles). This is the reason why CPM values cannot be compared to SPM values, since the same score for CPM falls in another IQ zone for a child aged 8 and 10 years. To eliminate this, we created a unified category system, combining the upper and lower two categories of CPM into the extremely high and very low categories, and merging the very low and extremely low of the SPM levels into the very low category. For the purposes of the study, IQ zones are only meaningful because of descriptive statistical characteristics, fitting based on the raw score obtained on the intelligence test for the same age. For the sake of clarity, we have removed the intelligence test values from the population characteristics. In the Methods section, we added two other references about the Hungarian standards. We deleted the corresponding part of Table 1. because of its low relevance to our results.

7. Define ADHD in Tables and captions.

Response: Done.

8. In defining NAT, RER, ALER, and GER, designate which are numerators, denominators in these ratios.

Response: In Data analysis section we have designated the ratios.

9. On line 374, the hippocampi of patients with Tourette syndrome were larger than whose? TS+ADHD or controls?

Response: We added to the manuscript: than that of the controls

10. References 3 and 9 are the same article.

Response: Thank you for the remark. We thought these are two separate article, because of they have different title (A personal 35 year perspective on Gilles de la Tourette syndrome: prevalence, phenomenology, comorbidities, and coexistent psychopathologies, and A personal 35 year perspective on Gilles de la Tourette syndrome: assessment, investigations, and management) and different page numbers (Lancet Psychiatry. 2015 Jan;2(1):68-87. and Lancet Psychiatry. 2015 Jan;2(1):88-104.)

Reviewer #2: 

As is to be expected for this kind of work, the sample sizes are reasonable, but likely insufficient to take any differences attributable to medication into account. N=3 TS and N=6 TS+ADHD children were medicated and medication has previously been found to affect associative learning in both TS and ADHD. I’m not sure that the medication details have been provided and I can’t see that potentially confounded effects of medication have been discussed at all within the present manuscript.

Response: We described in detail the medication of the patient groups in Participants subsection of Methods. We analyzed the impact of medication on the performance in a new subsection in Results from three different aspects. These result support that the medication has no significant modulatory effects on the results. Thus the separate comparison of the performances of the entire three patient groups (TS, TS+ADHD, TS+OCD/ASD) with their matched control group we provided in the revised manuscript. These new results were discussed, too. 

If the lack of difference in performance between TS and TS+ADHD groups suggests that the effects shown in Figure 2 (differences in NAT and ALER) are TS-mediated, should the same profile not also be shown in the 3rd TS group (comorbid with ASD or OCD)?

Response: In the Tourette syndrome and other (OCD and autism spectrum disorder, termed Tourette syndrome + OCD/ASD) group ten of eleven children were able to complete the test (7 OCD and 3 ASD comorbid patient). We have made this new group and took into the analysis. Thus the number of the participants increased to 46 in patient and control groups, too. We also discussed the new results.

The third TS group don’t seem to be shown in the figures, nor are the data considered with and without the exclusion of participants on medication?

Response: We have shown the results of the third group (TS + OCD/ASD). We have reanalyzed the data with and without the exclusion of participants with medication.

These result support that the medication has no significant modulatory effects on the results. Thus the separate comparison of the performances of the entire three patient groups (TS, TS+ADHD, TS+OCD/ASD) with their matched control group we provided in the revised manuscript. The case number of the unmedicated TS+ADHD and TS+OCD/ASD groups would be under 10 and this calls into question the relevance of statistical analysis. However, we have performed the comparison of the unmedicated groups with their matched controls and we provide the results here in these tables. These results are basically same to the results of the entire (medicated and unmedicated) patient subgroups. Thus we have decided to publish the results of the entire TS populations.

 TS (n=18) TS control (n=18) Mann-Whitney rank test

NAT median: 79.5

range: 52-130 median: 60

range: 46-124 U=246,

p = 0.009

ALER median: 0.102

rage: 0.018-0.325 median: 0.083

range: 0-0.186 U=221,

p = 0.064

RER median: 0.056

range: 0-0.333 median: 0.083

range: 0-0.361 U=140,

p = 0.482

GER median: 0.083

range: 0-0.667 median: 0.167

range: 0-0.917 U=126

p = 0.252

Table 1 Performance of the unmedicated patient with Tourette syndrome group compared with matched healthy control group

TS means Tourette syndrome. NAT means the number of the necessary trials in the acquisition phase of the paradigm, ALER means the error ratios in the acquisition phase of the paradigm, RER means the error ratios in the retrieval and GER in the generalization parts of the test phase.

 TS+ADHD (n=9) TS+ADHD control (n=9) Mann-Whitney rank test

NAT median: 62

range: 42-202 median: 75

range: 56-97 U=47.5,

p = 0.565

ALER median: 0.097

rage: 0-0.287 median: 0.105

range: 0.07-0.143 U=737,

p = 0.480

RER median: 0.056

range: 0.028-0.222 median: 0.083

range: 0.028-0.472 U=35.5,

p = 0.687

GER median: 0.083

range: 0-0.5 median: 0.167

range: 0-0.917 U=28,

p = 0.280

Table 2 Performance of the unmedicated patient with Tourette syndrome + ADHD group compared with matched healthy control group

TS means Tourette syndrome, ADHD means Attention Deficit Hyperactivity Disorder. NAT means the number of the necessary trials in the acquisition phase of the paradigm, ALER means the error ratios in the acquisition phase of the paradigm, RER means the error ratios in the retrieval and GER in the generalization parts of the test phase.

 TS+OCD/ASD (n=7) TS+OCD/ASD control (n=7) Statistical test 

NAT median: 89

range: 46-169 median: 61

range: 49-84 Welch test t(9.22)=2.08,

p = 0.076

ALER median: 0.136

rage: 0.043-0.172 median: 0.072

range: 0.019-0.125 t test t(12)=2.16,

p = 0.052

RER median: 0.028

range: 0.028-0.222 median: 0.111

range: 0.028-0.250 Mann-Whitney test U=20.5,

p = 0.645

GER median: 0.167

range: 0-0.583 median: 0.250

range: 0-0.750 t test t(12)=-0.822,

p = 0.427

Table 3 Performance of the unmedicated patient with Tourette syndrome + OCD/ASD group compared with matched healthy control group

TS means Tourette syndrome, OCD/ASD means the third patient group with OCD and ASD comorbidity. NAT means the number of the necessary trials in the acquisition phase of the paradigm, ALER means the error ratios in the acquisition phase of the paradigm, RER means the error ratios in the retrieval and GER in the generalization parts of the test phase.

Ln 123-128 - I didn’t understand the rationale to categorize IQ scores: like age, IQ is a continuous variable and to categorize the participants’ scores into IQ ranges may result in a loss of statistical power. Please explain. It’s also not clear how we are to understand the categories which run from ‘extremely high’ through to ‘very low’ with no further definition of these categories.

Response: In the study, we used the Standard Progressive Matrices (SPM) at volunteers over 12 years of age and the Colored Progressive Matrices (CPM) below the age of 12 years. The Hungarian standard values of the two tests cannot be completely matched. The SPM values are in age-independent categories (7 levels), while the CPM values are in the Hungarian standard for age-dependent categories (8 levels, over 95, 90-95, 75-90, 50-75, 25-50, 10-25, 5-10, and bellow 5 centiles). This is the reason why CPM values cannot be compared to SPM values, since the same score for CPM falls in another IQ zone for a child aged 8 and 10 years. To eliminate this, we created a unified category system, combining the upper and lower two categories of CPM into the extremely high and very low categories, and merging the very low and extremely low of the SPM levels into the very low category we use. For the purposes of the study, IQ zones are only meaningful because of descriptive statistical characteristics, fitting based on the raw score obtained on the intelligence test for the same age. For the sake of clarity, we have removed the intelligence test values from the population characteristics. In the Methods section, we added two other references about the Hungarian standards. In manuscript we deleted the connected part of Table 1. because of its low relevance to our results.

Ln 361-365 – I can’t see how this conclusion about the compensatory effect of ADHD follows. I thought the TS and TS+ADHD groups were not significantly different from each other? Now we’re getting a different comparison (each of the sub-groups with the controls) and the TS-ADHD group also happens to be smaller (N=15). Plus medication does not seem to have been taken into account.

Response: We have deleted the speculations about the primary effect of the TS or the ADHD in the alterations of learning functions of the patients. We have modified the abstract accordingly. We have described in the discussion only, which seems to be clear based on our results that not the ADHD is alone/primarily responsible for the associative learning disabilities of the entire TS patient group.

The writing is mostly understandable but the manuscript would benefit from further editing by a fluent English speaker, this is not really the job of the reviewer. For example, the Abstract which is most visible (and all some will read) could be better written (some specific suggestions below).

Response: Fluent English proofreading was done (see the attached certificate).

Ln 21-22 - The first sentence is a little awkward.

Ln22-23 – ‘the majority of the cognitive functions’ - ‘the majority of cognitive functions’

Ln 23 - ‘only little evidence’ – suggest rephrase

Ln 26 – ‘The acquired equivalence learning…’ - ‘Acquired equivalence learning…’

Ln 32 – ‘the entire patient group’ – please be more specific, presumably TS+ADHD

Ln 33 – ‘associations with lower effectiveness’ – please rephrase

Ln 36-37 – ‘parts of the test phase’ – plural so ‘depend…’ (and sentence seems to be missing a comma after hippocampus)

Response: We accepted all of these suggestions and connected the manuscript accordingly.

Minor comments and typos

Ln 140 - iOS – should be defined?

Response: Done.

Ln 184 – misplaced (and uninformative) figure caption text.

Response: We have inserted this figure caption text immediately after the first paragraph in which the figure is cited according to the instruction for authors.

Ln 211-212 – Data availability statement and link will need updating. I’m not sure I see the point of providing the data for peer review only at the first revision if the data have not been provided at this point (for peer review prior to the point of being on a ‘revise and resubmit’ ticket)? Elsewhere (in the additional information boxes) it says all relevant data are within the manuscript and its Supporting Information files but I can’t see that the data has in fact been provided at this point. The Supporting Information files seem to be figures.

Response: We added in the supporting information an excel table about the psychophysical performances of 46 TS patients and the 46 matched controls, which were used in the present study (S1 Table).

Ln 238-248 – I think this para is the figure legend.

Ln 272-277 – I think this para is the figure legend.

Ln 297-302 – I think this para is the figure legend (ditto for other figures, why is the legend in the text?)

Response: According to the instruction for authors we had to put the figure captions within the text.

Ln 579 – please make the figure caption self-explanatory rather than referring the reader back to the text.

Response: We added the detailed explanation to each figure.

Ln 583 - are the data for TS and TS+ADHD participants in fact shown separately in Figure 2? (As implied by the figure header.) If we’re looking at all the TS data relative to matched controls why not also include TS+OCD/ASD?

Response: We have prepared a new figure including the data of TS+OCD/ASD group and its matched healthy control group.

Ln 608 - ‘syndrom’ (typo)

Response: Done, as well as in the caption of Figure 5.

Ln 612 - please make the figure caption self-explanatory rather than referring the reader back to an earlier figure legend.

Response: We added the detailed explanation to each figure.

---

## [Decision Letter · Decision Letter 1]

14 May 2020

PONE-D-19-30158R1

Impairment of visually guided associative learning in children with Tourette syndrome

PLOS ONE

Dear Dr Eördegh,

Thank you for submitting your manuscript to PLOS ONE. After careful consideration, we feel that it has merit but does not fully meet PLOS ONE’s publication criteria as it currently stands. Therefore, we invite you to submit a revised version of the manuscript that addresses the points raised during the review process.

We would appreciate receiving your revised manuscript by Jun 28 2020 11:59PM. To enhance the reproducibility of your results, we recommend that if applicable you deposit your laboratory protocols in protocols.io, where a protocol can be assigned its own identifier (DOI) such that it can be cited independently in the future. For instructions see: http://journals.plos.org/plosone/s/submission-guidelines#loc-laboratory-protocols

We look forward to receiving your revised manuscript.

Kind regards,

Alexandra Kavushansky, PhD

Academic Editor

PLOS ONE

Additional Editor Comments (if provided):

Thank you for the corrections performed to the original version of the submission. Most of the issues raised by the Reviewers were addressed; there is still a question of the statistical analyses (e.g. correction for multiple comparisons, when comparing different subgroups (with/no medication, without/with different comorbidities) with their controls), making it hard to solidly base the conclusions on the obtained results.

Reviewers' comments:

Reviewer's Responses to Questions

**Comments to the Author**

1. If the authors have adequately addressed your comments raised in a previous round of review and you feel that this manuscript is now acceptable for publication, you may indicate that here to bypass the “Comments to the Author” section, enter your conflict of interest statement in the “Confidential to Editor” section, and submit your "Accept" recommendation.

Reviewer #1: (No Response)

2. Is the manuscript technically sound, and do the data support the conclusions?

Reviewer #1: No

3. Has the statistical analysis been performed appropriately and rigorously? 

Reviewer #1: No

4. Have the authors made all data underlying the findings in their manuscript fully available?

Reviewer #1: No

5. Is the manuscript presented in an intelligible fashion and written in standard English?

Reviewer #1: Yes

6. Review Comments to the Author

Reviewer #1: Thank you for the substantial revisions in response to Reviewer comments. Several additional comments are below.

Major Comments:

1. Does age differ significantly among subgroups? For example, is the nearly one-year age difference between the TS+ADHD and TS+OCD/ASD groups (patients or controls) significant?

2. Does tic severity differ significantly among patient subgroups? Increased tic severity can accompany comorbid conditions.

3. Does IQ differ significantly among patient subgroups? Of note, tic severity and IQ do not need to be included in Table 1, but a statement can be made that they do not differ among subgroups, if that is the case.

4. Does performance differ significantly among control subgroups? A pitfall of enrolling one control per patient (rather than the same 10-20 controls used for all subgroups) is that the control subgroups may differ among themselves. Whereas age and gender appear to match, task performance varies. Specifically, the TS+ADHD controls had a median NAT of 75, whereas the other control subgroups had median NATs of 60.5 and 60, respectively. The TS+ADHD controls had a median ALER of 0.105, whereas the other control subgroups both had median ALER scores of 0.083. This potentially significant difference among control subgroups is critical, for it could be the reason the TS+ADHD patient group does not differ from controls. This would change the interpretation of the data from ADHD not altering cognitive functions, to indeed altering visual acquisition.

Minor Comments:

1. On line 193, what is meant by “involuntary…attention”? Involuntary attention is to a unattended, unexpected stimulus that reflexively draws the subject’s gaze. “Undivided attention” sounds more like voluntary (albeit undivided) attention.

2. The task description in the Methods is far more clear. Thank you.

3. On line 233, reverse the order of words to read “with and without.”

4. On line 306, please delete the word “After.”

5. In Tables 2 and 3, add an “n” in “range” for the ALER of patients with TS.

6. On line 369, what alterations are referenced? Perhaps “significant findings”?

7. In the Results, please place the data of patients with TS+ADHD before that of TS+OCD to mirror all preceding text and tables.

7. PLOS authors have the option to publish the peer review history of their article (what does this mean?). If published, this will include your full peer review and any attached files.

Reviewer #1: Yes: Cameron B. Jeter

---

## [Author Response · Author response to Decision Letter 1]

21 May 2020

First of all, we would like to express our gratitude for the scholarly and highly helpful criticism of the Academic Editor and the Referees, which helped us to improve the quality of our study. We accepted all of the suggestions made changes accordingly. The suggestions are answered below itemized; textual changes are marked in the final re-revised manuscript.

Reviewer #1: Thank you for the substantial revisions in response to Reviewer comments. Several additional comments are below.

We can’t understand the concern of the reviewer about the data availability, because we have uploaded in the previous (revised) draft of the manuscript the entire xls table, which contains the psychophysical performances of all patients and controls in the supplementary materials of the manuscript.

Major Comments:

1. Does age differ significantly among subgroups? For example, is the nearly one-year age difference between the TS+ADHD and TS+OCD/ASD groups (patients or controls) significant?

Response: The age was not different among the TS patient subgroups. We have added this statement to the manuscript. The age was not different among the control subgroups, too.

2. Does tic severity differ significantly among patient subgroups? Increased tic severity can accompany comorbid conditions.

Response: The tic severity was not different among the TS patient subgroups. We have added this statement to the manuscript. 

3. Does IQ differ significantly among patient subgroups? Of note, tic severity and IQ do not need to be included in Table 1, but a statement can be made that they do not differ among subgroups, if that is the case.

Response: The IQ was not different among the TS patient subgroups. We have added this statement to the manuscript. The IQ was not different among the control subgroups, too.

4. Does performance differ significantly among control subgroups? A pitfall of enrolling one control per patient (rather than the same 10-20 controls used for all subgroups) is that the control subgroups may differ among themselves. Whereas age and gender appear to match, task performance varies. Specifically, the TS+ADHD controls had a median NAT of 75, whereas the other control subgroups had median NATs of 60.5 and 60, respectively. The TS+ADHD controls had a median ALER of 0.105, whereas the other control subgroups both had median ALER scores of 0.083. This potentially significant difference among control subgroups is critical, for it could be the reason the TS+ADHD patient group does not differ from controls. This would change the interpretation of the data from ADHD not altering cognitive functions, to indeed altering visual acquisition.

Response: Thank You for this important question. None of the performances (NAT, ALEG, RER, GER) were different among the control subgroups. We added these results to the manuscript. 

Minor Comments:

1. On line 193, what is meant by “involuntary…attention”? Involuntary attention is to a unattended, unexpected stimulus that reflexively draws the subject’s gaze. “Undivided attention” sounds more like voluntary (albeit undivided) attention.

2. The task description in the Methods is far more clear. Thank you.

3. On line 233, reverse the order of words to read “with and without.”

4. On line 306, please delete the word “After.”

5. In Tables 2 and 3, add an “n” in “range” for the ALER of patients with TS.

6. On line 369, what alterations are referenced? Perhaps “significant findings”?

7. In the Results, please place the data of patients with TS+ADHD before that of TS+OCD to mirror all preceding text and tables.

Answer: We accepted all of these minor suggestions and corrected the manuscript accordingly.

---

## [Decision Letter · Decision Letter 2]

2 Jun 2020

Impairment of visually guided associative learning in children with Tourette syndrome

PONE-D-19-30158R2

Dear Dr. Eördegh,

We’re pleased to inform you that your manuscript has been judged scientifically suitable for publication and will be formally accepted for publication once it meets all outstanding technical requirements.

Kind regards,

Alexandra Kavushansky, PhD

Academic Editor

PLOS ONE

Additional Editor Comments (optional):

Reviewers' comments:

Reviewer's Responses to Questions

**Comments to the Author**

1. If the authors have adequately addressed your comments raised in a previous round of review and you feel that this manuscript is now acceptable for publication, you may indicate that here to bypass the “Comments to the Author” section, enter your conflict of interest statement in the “Confidential to Editor” section, and submit your "Accept" recommendation.

Reviewer #1: All comments have been addressed

2. Is the manuscript technically sound, and do the data support the conclusions?

Reviewer #1: Yes

3. Has the statistical analysis been performed appropriately and rigorously? 

Reviewer #1: Yes

4. Have the authors made all data underlying the findings in their manuscript fully available?

Reviewer #1: Yes

5. Is the manuscript presented in an intelligible fashion and written in standard English?

Reviewer #1: Yes

6. Review Comments to the Author

Reviewer #1: Thank you for addressing reviewer comments; the manuscript will make a nice publication.

I had not previously seen the embedded link in the manuscript PDF to your uploaded data. It is there. Thanks.

7. PLOS authors have the option to publish the peer review history of their article (what does this mean?). If published, this will include your full peer review and any attached files.

Reviewer #1: No

---

## [Editor Report · Acceptance letter]

5 Jun 2020

PONE-D-19-30158R2 

Impairment of visually guided associative learning in children with Tourette syndrome 

Dear Dr. Eördegh:

I'm pleased to inform you that your manuscript has been deemed suitable for publication in PLOS ONE. Congratulations! Your manuscript is now with our production department. 

Kind regards, 

on behalf of

Dr. Alexandra Kavushansky 

Academic Editor

PLOS ONE